Multichannel speech enhancement for automatic speech recognition: a literature review

Zaland Zubair 1
Mustafa Mumtaz Begum 1 mumtaz@um.edu.my
http://orcid.org/0000-0002-1240-5406 Mat Kiah Miss Laiha 2
Ting Hua-Nong 3
Mohamed Yusoof Mansoor Ali 4
Mohd Don Zuraidah 5
http://orcid.org/0000-0002-0684-703X Muthaiyah Saravanan 6
1 Department of Software Engineering, Faculty of Computer Science and Information Technology, Universiti Malaya , Kuala Lumpur , Malaysia
2 Department of Computer System and Technology, Faculty of Computer Science and Information Technology, Universiti Malaya , Kuala Lumpur , Malaysia
3 Department of Biomedical Engineering, Faculty of Engineering, Universiti Malaya , Kuala Lumpur , Malaysia
4 Faculty of Business Finance and Information Technology, MAHSA University , Jenjarom, Selangor , Malaysia
5 Institute of Languages, UCSI University , Kuala Lumpur , Malaysia
6 School of Business and Technology, International Medical University , Kuala Lumpur , Malaysia
Alatas Bilal
Electronic publication date: 2025 Mar 27
Publication date: 2025
Volume: 11
Electronic Location ID: e2772
Received 2024 Oct 4; Accepted 2025 Feb 26
Copyright: © 2025 Zaland et al.
Copyright year: 2025
Copyright holder: Zaland et al.
License: This is an open access article distributed under the terms of the Creative Commons Attribution License, which permits unrestricted use, distribution, reproduction and adaptation in any medium and for any purpose provided that it is properly attributed. For attribution, the original author(s), title, publication source (PeerJ Computer Science) and either DOI or URL of the article must be cited.
License URL: https://creativecommons.org/licenses/by/4.0/

Keywords: Automatic speech recognition, Noise suppression, Noise types, Multichannel speech enhancement, Systematic literature review

Funding: Ministry of Higher Education under the Fundamental Research Grant Scheme FRGS/1/2023/ICT09/UM/02/1 This research was financially supported by the Ministry of Higher Education under the Fundamental Research Grant Scheme (FRGS/1/2023/ICT09/UM/02/1). The funders had no role in study design, data collection and analysis, decision to publish, or preparation of the manuscript.

==============================
Multichannel speech enhancement (MCSE) is crucial for improving the robustness and accuracy of automatic speech recognition (ASR) systems. Due to the importance of ASR systems, extensive research has been conducted in MCSE, leading to rapid advancements in methods, models, and datasets. Most previous reviews point to the lack of a systematic literature review of MCSE for ASR systems. This systematic literature review aims to (1) perform a comprehensive review of the existing approaches in MCSE for ASR, (2) analyze the performance of the MCSE and ASR for various techniques, models, as well as noise data and environments, and (3) discuss the challenges, limitations, and future research directions in this research area. We conducted keyword searches on several electronic databases such as Google Scholar, IEEE Xplore, ScienceDirect, SpringerLink, ACM Digital Library, and ISI Web of Knowledge to identify relevant journal and conference articles. We selected 240 articles based on inclusion criteria from the initial search results and ended with 35 experimental articles when exclusion criteria were applied. Through backward snowballing and the quality assessment, the final tally was 40 articles, comprising 23 journals, and 17 conference articles. The review shows that there is an increasing trend in MCSE for ASR with word error rate (WER), perceptual evaluation of speech quality (PESQ), and short-time objective intelligence (STOI) as common forms of performance measures. One of the major issues that we found in the review is the generality and comparability of the MCSE works, making it difficult to come up with unified solutions to noises in speech recognition. This systematic literature review has extensively examined MCSE and ASR techniques. Key findings include identifying MCSE methods that help ASR performance across various models, techniques, noise, and environments. We also identify several key areas researchers can explore in the future due to their promising potential.

Introduction

Multichannel speech enhancement (MCSE) for automatic speech recognition (ASR) systems utilizes multiple microphones to capture audio from different angles and distances. MCSE is an important area of research that aims to improve the quality of audio data captured by microphones for ASR systems (Kong et al., 2021). MCSE with ASR works as an independent group of joined modules, each taking the output from the subsequent module as input. A typical MCSE begins with speech signals from multiple microphones, which are initially fed to a beamformer. The beamformer aims to concentrate on a particular direction from which the desired signals are being transmitted. Beamformers also avoid signals from other sources and directions to minimize noise and interference. The beamforming output (now an enhanced signal) is then fed to ASR for speech recognition (Ochiai, Watanabe & Katagiri, 2017).

MCSE tackles the limitations of single-channel speech enhancement (SCSE) in complex and noisy environments, where SCSE is prone to introducing distortions that negatively affect enhanced speech quality (Xiao, Pennington & Schoenholz, 2020; Kinoshita et al., 2020). On top of that, SCSE’s expensive and slow computation results in difficulties in deploying the model (Wu et al., 2021) and cannot utilize spatial cues to separate overlapping speakers and sounds (Lin, Zhang & Qian, 2023).

MCSE performs better than SCSE in complex acoustic environments in a variety of background noises and reverberation (Lin, Zhang & Qian, 2023). It captures sounds from multiple directions, through which they can provide more consistent accuracy in scenarios where speaker position and proximity from the input device may vary (Das et al., 2021). Some of the MCSE applications of ASR include teleconferencing (Benesty, 2008), hearing aids (Gannot et al., 2017), voice-controlled systems utilizing speech and speaker recognition (Wang & Chen, 2018), voice-activated controls, wearable devices (Cherukuru, Mustafa & Subramaniam, 2022), and in-vehicle communication (Hu et al., 2006).

Researchers have conducted several surveys that examine the MCSE for ASR (Hermus, Wambacq & Van hamme, 2006; Haeb-Umbach et al., 2021; Yuliani et al., 2021; Zilvan et al., 2021; Park et al., 2022; Chhetri et al., 2023; Dua, Akanksha & Dua, 2023; Tanveer et al., 2023; Prabhavalkar et al., 2024). However, none of the existing articles are systematic literature reviews. Systematic reviews offer a clear and comprehensive overview of available evidence on MCSE speech enhancement for ASR. Adopting a systematic review protocol, using specific research questions and quality assessment enables the selection of articles relevant to MCSE approaches, models, datasets, and performance measures critical for ASR’s real-life applications.

This systematic literature review (SLR) aims to (1) provide a comprehensive review of the existing approaches in MCSE for ASR, (2) analyze the performance of the MCSE and ASR, and (3) discuss the challenges, limitations, and future research directions in this research area.

The remainder of the article is structured as follows: “Research Background” describes the background work related to this review including the existing review articles and their limitations. “Research Methods” explains the methods used for the SLR, including the formulation of research questions and the study’s selection criteria. “Results” details the findings of the research questions, while “Discussion” discusses the findings of the review and the challenges, limitations, and future directions in this research area. “Conclusions” concludes the review.

Research background

The presence of noise is very challenging for ASR systems as their performance depends on clear and noise-free speech signals. Noise is unavoidable in real-life applications, and it causes inaccuracies in transcription and a negative user experience when using ASR systems. Noise suppression is crucial for improving audio quality and enhancing speech recognition quality. Many works have been conducted to tackle the issue of noise in ASR recognition. MCSE is an important area of research that aims to improve the quality of audio data captured by microphones for ASR systems (Lin, Zhang & Qian, 2023).

With the growing number of publications, researchers offer many different solutions, simulating different environments in their experiments by adding noise to the speech data in various manners, to improve the performance of ASR systems in noisy environments. However, no systematic review has been conducted that summarizes and synthesizes the current solutions of MCSE for ASR in various types of noises that may be encountered in real life, as well as evaluating the current performance of MCSE that provides the understanding of the effectiveness of the current solutions.

Table 1 provides the areas of focus of the existing survey articles, including MCSE, ASR system, speech datasets, and performance measures. All the existing reviews of MCSE are survey types that lack systematic methodology, such as establishing research questions, inclusion/exclusion criteria, and quality assessment.

Table 1 Analysis of previous survey articles.

Article	Review approach	Years covered	Digital repository	Quality assessment	MCSE techniques	Noises	Performance measures	
Hermus, Wambacq & Van hamme (2006)	Survey	X	X	No	X	✓	✓	
Zilvan et al. (2021)	Survey	X	X	No	✓	✓	✓	
Prabhavalkar et al. (2024)	Survey	Recent 14 years	X	No	X	X	✓	
Park et al. (2022)	Survey	Recent 20 years	X	No	✓	X	✓	
Tanveer et al. (2023)	Survey	X	X	No	✓	X	✓	
Yuliani et al. (2021)	Survey	Recent 8 years	X	No	✓	✓	✓	
Chhetri et al. (2023)	Survey	X	X	No	✓	X	✓	
Dua, Akanksha & Dua (2023)	Survey	X	X	No	X	✓	✓	
Haeb-Umbach et al. (2021)	Survey	X	X	No	✓	✓	✓	

The rationale of the review

MCSE is an emerging field of research that assists in the improvement of the recognition accuracy of ASR systems for real-world applications. ASR is used in all speech-based commands and machine interaction in areas like customer relations, autonomous machines, AI, and smart homes. MCSE is critical for real-life applications of ASR systems. With the increasing interest in MCSE, several previous articles have attempted to summarize the various methods used for MCSE, the existing surveys inadequately address critical aspects such as resources including datasets and noise types, as well as the current performance for gauging the effectiveness of MCSE. This SLR provides a comprehensive analysis of the MCSE techniques and resources on the performance of ASR systems. It also helps explore the gaps in current research and improve our understanding of the field. The objective of our work is to provide researchers and developers with the foundational knowledge of current advancements and limitations in the field of MCSE for ASR.

Research methods

This SLR of MCSE for the ASR system was carried out according to the Preferred Reporting Items for SLRs and Meta-Analyses (PRISMA) reporting guidelines. The quality of the SLR can vary greatly, and it is important to ensure that the SLR is conducted in a rigorous and systematic manner (Kitchenham, 2004; Alharbi et al., 2021). As the Prisma checklist is not a methodological framework, but a series of suggestions or better recommendations to be implemented for the sound development of SLR, the methodological protocol by Kitchenham (2004) is adopted in this review.

Research questions

This article conducted an SLR of MCSE for ASR by formulating and addressing the following research questions (RQs):

RQ1- What are the existing approaches in multichannel speech enhancement for automatic speech recognition?

The motivation for RQ1 is to gain a clear understanding of existing research on the topic of ‘Flexible Multichannel Speech Enhancement for Noise Robust Frontend MCSE for ASR systems that covers the methods, noise datasets, and performance measures.

RQ2- What is the current performance of multichannel speech enhancement and automatic speech recognition in the existing studies?

This research question examines the performance of MCSE and ASR systems as reported in existing research.

RQ3- What are the challenges, limitations, and future research directions raised by the existing studies in multichannel speech enhancement for automatic speech recognition?

This research question identifies the challenges, limitations of existing works and methods, and future directions.

Search strategy

We selected the following databases: GoogleScholar, IEEE Xplore, ScienceDirect, SpringerLink, ACM Digital Library, and ISI Web of Knowledge to identify the relevant articles. The articles were identified using a set of search queries using Boolean operators, as common in the literature. The following are the search queries used in this research.

(“Multichannel speech enhancement” OR “MCSE” AND (“ASR” OR “Automatic Speech Recognition”) AND (“Noise” OR “Noisy environment” OR “Noise suppression” OR “Noise reduction” OR “Multichannel noise suppression” OR “Noise filtering”).

Based on the search string above, many article titles were extracted for initial screening. Two researchers (ZZ and MBM) separately screened the titles and abstracts of the studies based on a set of inclusion criteria, and any disagreement between the researchers was solved by discussion with the third researcher (MAY). Such selection procedures reduce the biases of selection as it involves several researchers.

After the initial selection, a series of exclusion criteria were applied to remove articles that may not fit this research such as article types and content coverage. A backward snowballing was applied after the exclusion criteria screening to identify relevant articles that may be overlooked during the keyword search. Finally, a quality assessment was conducted on the screened articles to determine the quality of the contents that meet the purpose of this review.

The following inclusion criteria (IC) and exclusion criteria (EC) were applied to select the most relevant articles from the search results.

IC1 Studies written in English

IC2 Multichannel speech signals

IC3 ASR with Speech Enhancement

IC4 ASR with Noise filtering/reduction/suppression

EC1 Survey or review articles

EC2 Image/visual-based methods

EC3 Speech identification

EC4 Speech emotion recognition

Quality assessment

We evaluate each article based on specific assessment questions, as shown in Table 2 (refer to S1 File: Quality Assessment Scores). The score is based on (0) No, (1) Partially, (2) Yes. Articles that score 50% or less (from 10 marks) are rejected.

Table 2 Quality assessment questions.

QA	Question	
Q1	Are the aims of the research clearly stated?	
Q2	Does the article report recent methods and techniques in multi-channel speech enhancement for ASR?	
Q3	Any solutions provided towards the formulated RQs?	
Q4	Does the article provide results for assessing the performance of multi-channel speech enhancement for ASR?	
Q5	Does the article mention challenges and future directions related to the multi-channel speech enhancement for ASR?	

Data extraction, analysis, and synthesis

Relevant data was extracted from each research article after the quality assessment. The content extracted included title, year, problem statements, methodology, performance metrics, datasets, noise types, results, limitations, and future work. Due to extensive data extraction and the heterogeneity in experimental design, datasets, and performance metrics, we grouped them with some common classifications for extraction and analysis.

The data had to be sorted and synthesized to extract meaningful information from the literature. The data was tagged based on MCSE methods, noise datasets, and performance measures.

Results

This section describes the outcomes related to the systematic study RQs discussed above. Publication and selection biases are a potential threat to validity in all SLRs, and we cannot exclude the possibility that some research studies were missed, resulting in reduced precision and the potential for bias. Therefore, we made significant efforts to find all eligible research articles. We believe that our work significantly contributes to the role of MCSE in the ASR system. Figure 1 PRISMA flow diagram demonstrates the selection of studies included in this review.

Figure 1 PRISMA flow diagram demonstrating this review’s selected studies.

Our initial search generated 2,051 results. After removing duplicate records and rejection based on title and language, 240 articles were selected. After applying exclusion criteria EC1–EC4, the final tally is 35 articles. We added eight articles through backward snowballing, bringing the total articles to 43, comprising 26 journals and 17 conference articles. From the quality assessment scores, we excluded three more articles. Article (Coro et al., 2021) did not fulfil four quality assessment questions on noise types and datasets. Jiang & Chen (2023) and Moritz, Kollmeier & Anemuller (2016) did not clearly explain MCSE and/or ASR techniques and were also missing details in several other areas. Incidentally, the 40 selected were published between 2012 and 2024, as shown in Fig. 2. From Fig. 2, the bulk of articles (35%) were published in 2023, indicating increasing interest in MCSE research in recent times.

Figure 2 Distribution of articles per publication year.

RQ1- What are the existing approaches in multichannel speech enhancement for automatic speech recognition?

Table 3 summarizes the findings for RQ1, presenting the progress in MCSE for ASR in relation to the speech enhancement methods, noise datasets, and performance measures used in the existing studies.

Table 3 Analysis of MCSE methods, noise datasets, and performance measures used in the existing studies.

	Domain	Methods	Technique	Articles	
Multichannel speech enhancement for ASR	Signal processing-based methods	Blind source separation (n = 5)	Source Splitting technique (Proposed)	Subramanian et al. (2022)	
Independent component Analysis	Liu et al. (2023)	
T-F masking	Kanda et al. (2023), Bu et al. (2022), Wang & Cavallaro (2021)	
Beamforming (n = 22)	Neural Beamformer	Chang et al. (2023), Olivieri et al. (2023), Ochiai, Watanabe & Katagiri (2017), Jukić, Balam & Ginsburg (2023), Shi et al. (2022), Wang & Cavallaro (2021)	
Minimum Variance Distortion-less Response (MVDR) Beamformer	Kanda et al. (2023), Meyer et al. (2016), Shimada et al. (2019), Subramanian et al. (2022), Park et al. (2023), Bu et al. (2022)	
Adaptive Beamformer	Chhetri et al. (2018)	
Super-directive beamformer	Sainath et al. (2017), Sadeghi, Sheikhzadeh & Emadi (2024), Li et al. (2021)	
Beamformer Filter Bank (BF-FBANK	Purushothaman, Sreeram & Ganapathy (2020)	
Generalized Eigenvalue (GEV) beamformer	Chai et al. (2022), Li et al. (2021)	
Beamforming	Cherukuru & Mustafa (2024), Liu et al. (2023), Jokisch et al. (2021)	
Delay and sum (DS) beamformer	Novoa et al. (2021), Jukić, Balam & Ginsburg (2023)	
Multichannel non-negative matrix factorization (n = 3)	MNMF	Shimada et al. (2019), Kim et al. (2012), Fang et al. (2023)	
Computational Auditory Scene Analysis (CASA) (n = 2)	Ideal Binary Mask (IBM	Bu et al. (2022)	
Maximum A Posteriori (MAP)	Yadava & Jayanna (2019)	
Dereverberation (n = 5)	Discrete Wavelet Transform (DWT		Cherukuru & Mustafa (2024)	
Weight Prediction Error (WPE)		Sadeghi, Sheikhzadeh & Emadi (2024), Chai et al. (2022), Park et al. (2023)	
Spatio Spectral Autoregressive Modeling (SSAM)		Purushothaman, Sreeram & Ganapathy (2020)	
Deep learning based methods	Neural network based (n = 18)	Convolutional Neural Network (CNN	Chang et al. (2023), Braun & Gamper (2022), Cherukuru & Mustafa (2024), Lin et al. (2023)	
Deep Neural Network (DNN)	Chhetri et al. (2018), Shi et al. (2022), Shukla, Shajin & Rajendran (2024), Velásquez-Martínez et al. (2023), Deepak et al. (2022), Martinez, Moritz & Meyer (2014), Moritz, Kollmeier & Anemuller (2016)	
Convolutional Long-short term Deep Neural Network (CLDNN)	Sainath et al. (2017)	
Spatial HuBERT	Dimitriadis et al. (2023)	
Deep Convolutional Neural Network	Pandey et al. (2021)	
U-Net	Jiang et al. (2023), Bu et al. (2022)	
Mask Scalar Prediction	Narayanan et al. (2023)	
			Neural Network Generalized Sidelobe Canceller	Li et al. (2021)	
	Noise type	Articles	
Noise Dataset	Colored noise (n = 3)	Chang et al. (2023), Shukla, Shajin & Rajendran (2024), Velásquez-Martínez et al. (2023)	
White noise (n = 7)	Meyer et al. (2016), Moritz, Kollmeier & Anemuller (2016), Dua, Aggarwal & Biswas (2019a, 2019b), Sadeghi, Sheikhzadeh & Emadi (2024), Shukla, Shajin & Rajendran (2024), Cherukuru & Mustafa (2024)	
Ego noise (n = 5)	Fang et al. (2023), Novoa et al. (2021), Jokisch et al. (2021), Wang & Cavallaro (2021), Liu et al. (2023)	
Environmental noise (n = 33)	Chang et al. (2023), Narayanan et al. (2023), Kanda et al. (2023), Meyer et al. (2016), Chhetri et al. (2018), Sainath et al. (2017), Olivieri et al. (2023), Dimitriadis et al. (2023), Shimada et al. (2019), Purushothaman, Sreeram & Ganapathy (2020), Braun & Gamper (2022), Ochiai, Watanabe & Katagiri (2017), Martinez, Moritz & Meyer (2014), Jukić, Balam & Ginsburg (2023), Pandey et al. (2021), Jiang et al. (2023), Sadeghi, Sheikhzadeh & Emadi (2024), Chai et al. (2022), Li et al. (2021), Subramanian et al. (2022), Deepak et al. (2022), Qu, Weber & Wermter (2023), Shukla, Shajin & Rajendran (2024), Cherukuru & Mustafa (2024), Velásquez-Martínez et al. (2023), Lin et al. (2023), Yadava & Jayanna (2019), Bu et al. (2022), Fang et al. (2023), Novoa et al. (2021), Jokisch et al. (2021), Wang & Cavallaro (2021), Dua, Aggarwal & Biswas (2019b)	
Reverberation (n = 16	Chang et al. (2023), Chhetri et al. (2018), Sainath et al. (2017), Olivieri et al. (2023), Dimitriadis et al. (2023), Braun & Gamper (2022), Moritz, Kollmeier & Anemuller (2016), Jukić, Balam & Ginsburg (2023), Sadeghi, Sheikhzadeh & Emadi (2024), Chai et al. (2022), Li et al. (2021), Qu, Weber & Wermter (2023), Shi et al. (2022), Cherukuru & Mustafa (2024), Park et al. (2023), Novoa et al. (2021)	
	TYPE	Metric	Articles	
Performance measures	ASR	WER (n = 27)	Chang et al. (2023), Narayanan et al. (2023), Kanda et al. (2023), Meyer et al. (2016), Chhetri et al. (2018), Sainath et al. (2017), Shimada et al. (2019), Purushothaman, Sreeram & Ganapathy (2020), Braun & Gamper (2022), Moritz, Kollmeier & Anemuller (2016), Martinez, Moritz & Meyer (2014), Jukić, Balam & Ginsburg (2023), Jiang et al. (2023), Sadeghi, Sheikhzadeh & Emadi (2024), Chai et al. (2022), Subramanian et al. (2022), Deepak et al. (2022), Qu, Weber & Wermter (2023), Huang et al. (2022), Shi et al. (2022)), Cherukuru & Mustafa (2024), Park et al. (2023), Velásquez-Martínez et al. (2023), Bu et al. (2022), Fang et al. (2023), Novoa et al. (2021)	
WRR (n = 3)	Dua, Aggarwal & Biswas (2019a), Cherukuru & Mustafa (2024), Jokisch et al. (2021)	
CER (n = 4)	Jiang et al. (2023), Li et al. (2021), Ochiai, Watanabe & Katagiri (2017), Shi et al. (2022)	
PER (n = 1)	Dimitriadis et al. (2023)	
SER (n = 1)	Park et al. (2023)	
FRR (n = 1)	Chhetri et al. (2018)	
MCSE	PESQ (n = 10)	Olivieri et al. (2023), Shimada et al. (2019), Ochiai, Watanabe & Katagiri (2017), Sadeghi, Sheikhzadeh & Emadi (2024), Shukla, Shajin & Rajendran (2024), Velásquez-Martínez et al. (2023), Lin et al. (2023), Yadava & Jayanna (2019), Bu et al. (2022), Wang & Cavallaro (2021), Liu et al. (2023)	
STOI (n = 5)	Shukla, Shajin & Rajendran (2024), Velásquez-Martínez et al. (2023), Lin et al. (2023), Bu et al. (2022), Liu et al. (2023)	
ESTOI (n = 2)	Olivieri et al. (2023), Sadeghi, Sheikhzadeh & Emadi (2024)	
MSE (n = 2)	Shukla, Shajin & Rajendran (2024), Velásquez-Martínez et al. (2023)	
SDR (n = 3)	Olivieri et al. (2023), Ochiai, Watanabe & Katagiri (2017), Shi et al. (2022)	
SAR (n = 1)	Olivieri et al. (2023)	
SI-SNR (n = 1)	Braun & Gamper (2022)	
MPE (n = 1)	Dua, Aggarwal & Biswas (2019b)	
SIR (n = 1)	Kim et al. (2012)	
POLQA (n = 1)	Fang et al. (2023)	

Multichannel speech enhancement

Multichannel speech enhancement was divided into four major techniques including blind source separation (BSS), beamforming, multichannel non-negative matrix factorization (NMF), and computational auditory scene analysis (CASA). BSS techniques were opted for by five articles, which perform noise suppression by extracting individual source signals from a mixture without requiring detailed prior knowledge about the sources, its main objective being to recover the original signals by using the statistical properties of the mixed signals. Key techniques within BSS include independent component analysis (ICA) (Liu et al., 2023) and time-frequency (T-F) masking (Wang & Cavallaro, 2021; Bu et al., 2022; Kanda et al., 2023). Figure 3 shows the process of single and multichannel BSS, for the multichannel the processes involved in different microphone and source settings are defined. For example, if the number of sources is less than the number of microphones then dimension reduction takes place followed by one of the BSS techniques such as ICA and Multichannel-NMF jointly known as independent low-rank matrix analysis (ILRMA) (Sawada et al., 2019).

Figure 3 MCSE techniques for blind source separation (Sawada et al., 2019).

Among the 40 selected articles, over half (n = 22) use beamforming. Beamforming performs directional signal transmission from multiple sensors using signal processing and precise and real-time channel state information (CSI) for beam customization (Sainath et al., 2017; Chhetri et al., 2018; Chang et al., 2023; Olivieri et al., 2023).

The beamforming can be divided into two basic types: fixed (Benesty, Chen & Huang, 2008) and adaptive beamforming (Gannot & Cohen, 2008) where fixed beamforming uses a static noise source and direction of speech (Cherukuru & Mustafa, 2024). Adaptive beamforming involves the directivity of the speech signal differing from the change in acoustic environments (Jokisch et al., 2021; Cherukuru & Mustafa, 2024). There are several types of beamformers utilized by researchers in various articles including neural beamformers (Xiao et al., 2016; Heymann, Drude & Haeb-Umbach, 2016; Erdogan et al., 2016; Ochiai, Watanabe & Katagiri, 2017; Purushothaman, Sreeram & Ganapathy, 2020; Wang & Cavallaro, 2021; Shi et al., 2022; Chang et al., 2023; Park et al., 2023; Liu et al., 2023), minimum variance distortionless response (MVDR) beamformers, (Sainath et al., 2017; Chhetri et al., 2018; Li et al., 2021; Bu et al., 2022; Chai et al., 2022; Subramanian et al., 2022; Park et al., 2023), delay and sum (DS) beamformers (Novoa et al., 2021; Jukić, Balam & Ginsburg, 2023), and super-directive beamformers (Olivieri et al., 2023). Figure 4 shows a generic beamformer for speech enhancement in an end-to-end ASR proposed by Ochiai, Watanabe & Katagiri (2017).

Figure 4 End-to-end ASR using beamforming (Ochiai, Watanabe & Katagiri, 2017).

Very few studies adopted the NMF methods for speech enhancement (Kim et al., 2012; Shimada et al., 2019; Fang et al., 2023). NMF effectively extracts meaningful features from a set of non-negative vectors of data, making it an effective method for dimensionality reduction. This approach helps reduce the data complexity while preserving the essential auditory features. The multichannel NMF combines spatial and spectral information effective for source separation in multichannel recordings (Ozerov, Févotte & Vincent, 2018). The popularity of NMF is affected due to the algorithm being prone to leaving residual noise when there is a large overlap between speech and noise in the frequency domain (Kim et al., 2012). Figure 5 shows the MNMF technique proposed by Shimada et al. (2019) estimating the steering vector and covariance matrix to optimize a beamformer.

Figure 5 MCSE technique using multichannel NMF (Shimada et al., 2019).

CASA-based techniques were utilized in two articles (Yadava & Jayanna, 2019; Bu et al., 2022). It is a field of MCSE that separates and analyzes sound (speech and noise) sources, aiming to mimic the human perception of hearing sounds. CASA employs several techniques such as pitch-based segregation and spatial cues to identify and separate cues from sound sources. Its applications include hearing aids, ASR, and environmental sound analysis. CASA-based systems struggle in environments with reverberation, overlapping sounds (Zeremdini, Ben Messaoud & Bouzid, 2015), and varying noises. Figure 6 shows a CASA technique proposed by Zeremdini, Ben Messaoud & Bouzid (2015). It is used in the processing of sound signals in conjunction with machine learning to extract required information, often referred to as “machine listening” (Parker & Dockray, 2023).

Figure 6 MCSE technique using CASA (Zeremdini, Ben Messaoud & Bouzid, 2015).

Dereverberation (Purushothaman, Sreeram & Ganapathy, 2020; Chai et al., 2022; Park et al., 2023; Sadeghi, Sheikhzadeh & Emadi, 2024; Cherukuru & Mustafa, 2024) is the process that employs one of the microphones in an MCSE system to compensate for the reverberation effect in the environment without affecting the target speech. Five articles incorporate dereverberation techniques like weight prediction error (WPE) (Chai et al., 2022; Park et al., 2023; Sadeghi, Sheikhzadeh & Emadi, 2024), Discrete Wavelet Transform (DWT) (Cherukuru & Mustafa, 2024), and reverberation robust SSAR (Purushothaman, Sreeram & Ganapathy, 2020) model.

A total of 18 articles have incorporated the deep learning-based MCSE methods including convolutional neural network (CNN) (Braun & Gamper, 2022; Chang et al., 2023; Lin et al., 2023; Cherukuru & Mustafa, 2024) and deep neural network (DNN), (Chhetri et al., 2018; Shimada et al., 2019; Purushothaman, Sreeram & Ganapathy, 2020; Li et al., 2021; Wang & Cavallaro, 2021; Braun & Gamper, 2022; Velásquez-Martínez et al., 2023; Shukla, Shajin & Rajendran, 2024) other neural network-based speech enhancement methods include generative adversarial network (GAN) (Velásquez-Martínez et al., 2023; Lin et al., 2023), U-Net (Braun & Gamper, 2022; Jiang et al., 2023), deep convolutional neural network (DCNN) (Pandey et al., 2021), dual-path recurrent neural network (DPRNN) and Conv-TasNet (Shi et al., 2022). DNN and CNN are used in static and dynamic data (Cherukuru & Mustafa, 2024).

Datasets

Speech enhancement and ASR rely heavily on datasets for training and testing. The datasets, which typically comprise audio files (clean speech and a variety of noises at a different signal-to-noise ratio (SNR) and their transcription in text format, are vital for the development of MCSE for ASR. Data diversity, size, potential privacy issues (Dua, Aggarwal & Biswas, 2019b; Lin et al., 2023), and availability are some issues researchers face.

Clean speech datasets include audio files without noises along with their respective transcripts. Some of the common clean speech datasets used in MCSE and ASR research included LibriSpeech (Huang et al., 2022; Jiang et al., 2023; Narayanan et al., 2023; Qu, Weber & Wermter, 2023; Dimitriadis et al., 2023; Cherukuru & Mustafa, 2024), TIMIT (Meyer et al., 2016; Moritz, Kollmeier & Anemuller, 2016; Dua, Aggarwal & Biswas, 2019b; Wang & Cavallaro, 2021; Deepak et al., 2022; Huang et al., 2022; Fang et al., 2023), WSJ0 (Subramanian et al., 2022; Shi et al., 2022; Jukić, Balam & Ginsburg, 2023), AISHELL (Jiang et al., 2023) and variants of AURORA (Martinez, Moritz & Meyer, 2014; Meyer et al., 2016; Moritz, Kollmeier & Anemuller, 2016; Dua, Aggarwal & Biswas, 2019b; Cherukuru & Mustafa, 2024). Noise datasets are a collection of recordings that consist of common noises that are mixed with clean speech datasets to generate noisy speech. There are a variety of noisy speech and noise datasets available, such as CHiME (Ochiai, Watanabe & Katagiri, 2017; Shimada et al., 2019; Bu et al., 2022; Chai et al., 2022; Park et al., 2023) and Coswara (Bhattacharya et al., 2023).

There are many types of noises currently used in MCSE. White noise refers to noise data with consistent power throughout the frequency band (Dua, Aggarwal & Biswas, 2019a), while colored noise refers to noise data with variable frequencies in a consistent pattern (resulting in a constant value of frequency mean and variance (Velásquez-Martínez et al., 2023; Narayanan et al., 2023; Shukla, Shajin & Rajendran, 2024).

Noise can be further classified as; environmental noise (a variety of noises generated from the surroundings of the system and application such as a factory environment or a restaurant) (Kim et al., 2012; Martinez, Moritz & Meyer, 2014; Meyer et al., 2016; Sainath et al., 2017; Ochiai, Watanabe & Katagiri, 2017; Chhetri et al., 2018; Shimada et al., 2019; Dua, Aggarwal & Biswas, 2019a, 2019b; Purushothaman, Sreeram & Ganapathy, 2020; Bu et al., 2022; Chai et al., 2022; Deepak et al., 2022; Braun & Gamper, 2022; Jiang et al., 2023; Velásquez-Martínez et al., 2023; Narayanan et al., 2023; Chang et al., 2023; Qu, Weber & Wermter, 2023; Fang et al., 2023; Olivieri et al., 2023; Park et al., 2023; Lin et al., 2023; Shukla, Shajin & Rajendran, 2024; Cherukuru & Mustafa, 2024), ego noise (originates from the system itself) (Jokisch et al., 2021; Novoa et al., 2021; Wang & Cavallaro, 2021; Fang et al., 2023; Liu et al., 2023), and reverberation (refers to the persistence of sound from the original signal reflected from the sound-reflective bodies until it decays) (Sainath et al., 2017; Chhetri et al., 2018; Li et al., 2021; Chai et al., 2022; Narayanan et al., 2023; Olivieri et al., 2023; Sadeghi, Sheikhzadeh & Emadi, 2024). Some of the datasets that utilize environmental noise are CHiME (Ochiai, Watanabe & Katagiri, 2017; Shimada et al., 2019; Bu et al., 2022; Chai et al., 2022; Park et al., 2023), Environmental Sound Classification (ESC-50), and Forest Sound Classification (FSC22) datasets. Ego noise poses a great challenge for MCSE-ASR systems because of it is proximity to the microphones in relation to the speech source (Novoa et al., 2021), its ever-changing nature of white and stationary noise and non-stationary noise (Wang & Cavallaro, 2021), and the fact that all types of noise can occur at the same time.

Recent studies showed an increased preference for white rather than colored noise, while only Velásquez-Martínez et al. (2023), Narayanan et al. (2023), and Shukla, Shajin & Rajendran (2024) opted for pink noise in their research. MCSE for ASR systems requires adequate data to train for all types of noises for improved performance. Most studies simulated their models in various environments, and a significant number also used ego noise comprising datasets that included motor noise and propeller noise in the case of voice-activated drones (Jokisch et al., 2021; Novoa et al., 2021; Wang & Cavallaro, 2021; Fang et al., 2023; Liu et al., 2023).

Table 4 depicts the major datasets applied in the selected articles. From Table 4, some of the more commonly used datasets for speech enhancement are CHiME (Ochiai, Watanabe & Katagiri, 2017; Shimada et al., 2019; Purushothaman, Sreeram & Ganapathy, 2020; Bu et al., 2022; Chai et al., 2022; Jukić, Balam & Ginsburg, 2023; Cherukuru & Mustafa, 2024) containing environmental and reverberant noise; the TIMIT dataset (Dua, Aggarwal & Biswas, 2019b; Wang & Cavallaro, 2021; Deepak et al., 2022; Narayanan et al., 2023; Fang et al., 2023) containing ego, environmental, and reverberant noise; and LibriSpeech (Meyer et al., 2016; Bu et al., 2022; Huang et al., 2022; Jiang et al., 2023; Velásquez-Martínez et al., 2023; Chang et al., 2023; Qu, Weber & Wermter, 2023; Dimitriadis et al., 2023; Jukić, Balam & Ginsburg, 2023) for environmental and reverberant noise. All speech datasets are in English and recorded at frequency of 16 or 8 kHz. It was also noticed that some researchers create their noisy datasets instead of using the existing datasets, such as Novoa et al. (2021) for environmental and ego noise, Sainath et al. (2017), Pandey et al. (2021), Chang et al. (2023), Qu, Weber & Wermter (2023) developed their own noise datasets, while Sainath et al. (2017) and Chhetri et al. (2018) developed their reverberant noise datasets.

Table 4 Major datasets for speech enhancement.

Dataset	Noise type	Type of data	Size	Language	Frequency (kHz)	Article	
AgriDrone	Env	Ego noise, Env noise, Noisy speech	735 words	German	Not mentioned	Jokisch et al. (2021)	
AISHELL	Env	Speech	10 k h.	English	44.1, 16	Jiang et al. (2023)	
ATC	Ego	Noisy speech	10 k h.	English	8	Jiang et al. (2023)	
AMI	Reverberant	Speech, Noisy speech	5,080 h.	English	16	Kanda et al. (2023), Huang et al. (2022)	
AS, AVQ	Ego	Ego noise	4,620 utterances	English	8	Wang & Cavallaro (2021)	
AURORA	Env, white, Gaussian, Reverberant	Noise, Noisy speech	3 k h.	English	16	Meyer et al. (2016) Moritz, Kollmeier & Anemuller (2016) Martinez, Moritz & Meyer (2014) Cherukuru & Mustafa (2024)	
CHiME	Env, Reverberant	Noisy speech	5,080 h.	English	16	Shimada et al. (2019) Purushothaman, Sreeram & Ganapathy (2020) Ochiai, Watanabe & Katagiri (2017), Jukić, Balam & Ginsburg (2023) Chai et al. (2022) Velásquez-Martínez et al. (2023), Bu et al. (2022), Moritz, Kollmeier & Anemuller (2016)	
DEMAND	Env	Noise		English	16	Velásquez-Martínez et al. (2023), Fang et al. (2023)	
DNS challenge	Reverberant	Speech, Noise and noisy speech	100 k	English	16	Dimitriadis et al. (2023), Braun & Gamper (2022)	
Hindi language database	Env	Speech	1,000 sentences	Hindi	16	Dua, Aggarwal & Biswas, 2019a), Dua, Aggarwal & Biswas, 2019b	
LibriSpeech	Env, Reverb	Speech	2 k h.	English	16	Narayanan et al. (2023), Velásquez-Martínez et al. (2023), Jukić, Balam & Ginsburg (2023), Jiang et al. (2023), Velásquez-Martínez et al. (2023), Huang et al. (2022), Cherukuru & Mustafa (2024), Velásquez-Martínez et al. (2023), Bu et al. (2022)	
Self-developed	Env, Ego	n/a	15.2 h.	English		Novoa et al. (2021)	
Ego	n/a	60 k h.	English	16	Chang et al. (2023), Sainath et al. (2017), Pandey et al. (2021), Qu, Weber & Wermter (2023)	
Reverberant	n/a	39 k utterances	English		Chhetri et al. (2018)	
TIMIT	Env, Ego reverberant	Speech	2.6 k h.	English	8/16	Meyer et al. (2016), Dua, Aggarwal & Biswas, 2019b), Deepak et al. (2022), Shukla, Shajin & Rajendran (2024), Velásquez-Martínez et al. (2023), Fang et al. (2023), Bu et al. (2022), Velásquez-Martínez et al. (2023), Lin et al. (2023), Shukla, Shajin & Rajendran (2024)	
REVERB
WSJ	Env, Reverberant	Reverberant speech,
Noisy overlapping speech	960 h.	English	16	Velásquez-Martínez et al. (2023), Moritz, Kollmeier & Anemuller (2016)	
WSJ
UrbanSound	Env, Reverberant	Noisy speech, Noise	200 h.	English	8/16	Ochiai, Watanabe & Katagiri (2017), Bu et al. (2022), Velásquez-Martínez et al. (2023), Lin et al. (2023), Shukla, Shajin & Rajendran (2024), Shi et al. (2022), Velásquez-Martínez et al. (2023)	
VCTK, VCTK-N	Ego	Speech	2 k h.	English	16	Liu et al. (2023)	
Note:

NM, Not mentioned; Env, Environmental Noise.

As shown in Table 4, there are many different types of datasets used by the researchers in MCSE research. Different datasets allow researchers to experiment with their solutions with more diversified speech and noises, increasing the depth of knowledge and understanding of speech enhancement. On the other hand, diverse dataset applications result in the difficulty of comparing results among the research in MCSE for ASR. The datasets vary in terms of SNR, size, as well as the recording frequencies.

The selections of the datasets depend not only on the nature of the study but also on other factors, such as resource availability and constraints. For example, studies with better funding will likely develop their own set of datasets or access to purchased datasets, while others may opt for open or free access databases such as LibriSpeech or TIMIT (Meyer et al., 2016; Moritz, Kollmeier & Anemuller, 2016; Yadava & Jayanna, 2019; Dua, Aggarwal & Biswas, 2019b; Wang & Cavallaro, 2021; Deepak et al., 2022; Huang et al., 2022; Jiang et al., 2023; Velásquez-Martínez et al., 2023; Narayanan et al., 2023; Qu, Weber & Wermter, 2023; Fang et al., 2023; Dimitriadis et al., 2023; Shukla, Shajin & Rajendran, 2024; Cherukuru & Mustafa, 2024), making them as more common datasets.

Performance measure

In MCSE for ASR, the performance measures can be at MCSE or ASR output. Performance measures based on ASR include word error rate (WER), character error rate (CER), phoneme error rate (PER), word recognition rate (WRR), sentence error rate (SER),false rejection rate (FRR), recall rate of out of vocabulary (Recall ROOV). On the other hand, the performance metrics used for speech enhancement are perceptual evaluation of speech quality (PESQ) (Rix et al., 2002), short-time objective intelligence (STOI) (Taal et al., 2010), extended-STOI (Extended-STOI), signal-to-artifact ratio (SAR), signal-to-distortion ratio (SDR), scale-invariant signal to noise ratio (SI-SNR), mean perceptual error (MPE), mean squared error (MSE), signal-to-interference ratio (SIR), and perceptual objective listening quality analysis (POLQA). Metrics like WER, CER, and SER are not dependent on their corresponding clean speech signals. In contrast, metrics like PESQ, STOI, and ESTOI depend on the availability of reference signals (clean speech) for comparison making them dependent on simulation data.

WER is the percentage of recognition errors resulting from substitutions, deletions, and insertions over the total words spoken. Lower WER indicates good performance of the ASR system. PESQ is an objective measurement used to assess speech quality, which is a full-reference algorithm that analyzes the speech signal sample-by-sample after a temporal alignment of corresponding reference excerpts and test signals. PESQ results principally model mean opinion scores (MOS) that cover a scale from 1 (bad) to 5 (excellent). On the other hand, STOI is a method that evaluates the intelligibility of the speech signal particularly in noisy environments and its values range between 0 (bad) and 1 (excellent). STOI compares short-time spectral envelopes of degraded speech with clean speech and creates a correlation coefficient between the two. STOI measures the intelligibility of the processed input signal by comparing it with the clean reference signal, where higher value for the metric corresponds to a more intelligible speech signal (Bu et al., 2022; Velásquez-Martínez et al., 2023; Lin et al., 2023; Shukla, Shajin & Rajendran, 2024).

The lowest reported WER is 1.85% (Deepak et al., 2022), while the highest was 80% (Fang et al., 2023). The average WER among the selected articles is 20.13%, which means that, on average, the system will wrongly recognize one of the five inputs.

The lowest reported value for PESQ was 0.84 (Sadeghi, Sheikhzadeh & Emadi, 2024), while the highest value was reported to be 4.51 (Kim et al., 2012). The average PESQ among the selected articles is 2.35. The lowest reported value for STOI was 0.57 (Pandey et al., 2021), while the highest value was reported to be 0.98 (Shukla, Shajin & Rajendran, 2024). The average PESQ among the selected articles is 2.35.

RQ2- What is the current performance of multichannel speech enhancement and automatic speech recognition in the existing studies?

The analysis of the performance of the existing works is based on the SNR level, which is an indicator of the noise strength in speech input. The results from each article were extracted by taking the average of their scores across the range of SNR levels used by the article. To provide a better understanding of the results, the scores from each article include the ranges of SNR used by the researchers as shown in Figs. 7–9. However, in addition to SNR, this article assessed the performance scores for each metric in terms of evaluation data, to compare the results of different techniques on the same datasets. This section compares the results of 34 articles that used PESQ, STOI, and WER as performance metrics. Figure 7 shows the average WER scores of articles using similar datasets for evaluation.

Figure 7 Average word error rate for MCSE and ASR studies.

Figure 8 Average PESQ for MCSE and ASR studies.

Figure 9 Average STOI for MCSE and ASR studies.

The Aurora database (Nakamura et al., 2004) was opted by two studies, Cherukuru & Mustafa (2024) using CNN for noise suppression and Discrete Wavelet Transform (DWT) for dereverberation in a reverberant environment with various environmental background noise scored an average WER of 26.18. The article used the SNR range of −10 to 15 dB, with 5 dB steps, with the highest score of 30.16 at SNR = −10 dB and the lowest WER score of 21.92 at SNR = 0 dB. On the other hand, Martinez, Moritz & Meyer (2014) using the amplitude modulation filter bank (AMFB) with DNN scored a lower WER average of 13.6 with SNR ranging between 10 and 20 dB.

The CHiME database (Barker et al., 2015) was opted by six articles. Park et al. (2023) reported the highest average WER of 38.81% using the guided source separation with MVDR beamformer in a reverberant environment with Environmental background noises. Chai et al. (2022) used weight prediction error (WPE) and mask-based beamforming reporting an average WER of 34.75% when utilizing reverberant Environmental noise. Ochiai, Watanabe & Katagiri (2017) and Shimada et al. (2019) had similar WER scores of 12.12% and 11.93% respectively for environmental noise data with the former using Mel-scale filters with beamforming and multichannel NMF for noise estimation, and the latter adopting MVDR beamformer and Multichannel Wiener filter. Using environmental noise data on CNN for time-frequency mask estimation with MVDR Beamformer (Bu et al., 2022) scored the lowest average WER 8.45% for CHiME database at SNR = 0 dB.

For the LibriSpeech dataset (Panayotov et al., 2015; Cherukuru & Mustafa, 2024) scored the highest average WER of 31.66%, which was higher than the Aurora dataset for the same techniques. Velásquez-Martínez et al. (2023) using wavelet transform with DNN for speech recognition in noisy environments obtained average WER of 18.09% for SNR between −9 and 9 dB. Narayanan et al. (2023) used multichannel-cleanformer based on T-f masking and Acoustic echo cancelation (AEC) and achieved average WER of 14.73% for SNR between −10 and 5 dB. Jiang et al. (2023) with U-net and T5-error correction for Mandarin Air traffic-control ASR scored WER of 12.95%. The lowest average WER of 8.44% for the LibriSpeech dataset was obtained by Qu, Weber & Wermter (2023) using Multichannel Attention-based ASR with loss rescaling, followed closely with average WER 9.15% by Dimitriadis et al. (2023) using Masked prediction and denoising for Spatial HuBERT-based ASR with SNR between 0 and 20 dB.

The TIMIT dataset (Zue, Seneff & Glass, 1990) was opted by four articles, among them the lowest score was obtained by Deepak et al. (2022). The article used adaptive Tribonacci DNN scoring an average WER of 3.34%, the lowest average in this review article. On the other hand, the highest average WER of this review is also for the TIMIT database, where (Fang et al., 2023) using multichannel NMF with Wiener Filter and Variational Autoencoder (VAE) reported the highest WER of 59.54%.

The Wall Street Journal (WSJ) database (Paul & Baker, 1992) was opted by three studies among which the highest average WER was 20.68% by Subramanian et al. (2022) using MVDR beamformer for multi-source localization for SNR fixed at 0 and 30 dB. Jukić, Balam & Ginsburg (2023) scored a lower average WER of 14.12% using T-f masking for neural beamforming for SNR ranging between −5 and 20 dB. The best score for WSJ was obtained by Shi et al. (2022) scoring average WER of 9.50% using multi-source neural beamforming with the recurrent neural network (RNN) for SNR between 0 and 10 dB.

Other studies include datasets that are not used in more than one article, datasets developed in-house, and corpora in other languages. The highest average WER of 38.40% was reported in Li et al. (2021) using DNN with generalized sidelobe canceller (GSC) and Alpha-mini speech challenge (ASC) for SNR between 0 and 20 dB. Kim et al. (2012) generate WER of 31.25% using NMF when dealing with various environmental noises, while Chhetri et al. (2018) reported average WER od 30.16% using MCAEC and minimum variance distortion-less response (MVDR) beamformer in reverberant and ambient environmental noises. Using DNS challenge and AV speech datasets (Braun & Gamper, 2022), obtained average WER of 18.0% was obtained with Convolutional LSTM and DNN. Novoa et al. (2021) posted a WER of 13.76 using an MVDR beamforming technique for ego noise.

Using an In-house dataset with environmental noises, Chang et al. (2023) accomplished a low WER of 6.12% with MCAEC and neural beamforming. The best WER average was reported by Pandey et al. (2021) with WER 4.52% when employing recurrent neural network transducer (RNN-T) + deep convolutional recurrent network (DCRN) on Mel-FilterBank features for in-house dataset of real-world air traffic control electrical noise at SNR between −5 and 5 dB.

Figure 8 shows the average PESQ scores of the selected articles grouped according to their evaluation datasets. PESQ was used in research involving the CHiME dataset in two studies, among which Shimada et al. (2019) had a higher average PESQ score of 2.64, whereas Ochiai, Watanabe & Katagiri (2017) obtained an average PESQ score of 1.73. For LibriSpeech, Velásquez-Martínez et al. (2023) scored 1.82. Four articles used PESQ for the evaluation of the TIMIT database, where Yadava & Jayanna (2019) scored the highest PESQ score of 4.39. The next score was reported by Shukla, Shajin & Rajendran (2024) scored 1.97. Wang & Cavallaro (2021) using DNN-based beamforming posted a PESQ score of 1.67. The lowest score was obtained by Sadeghi, Sheikhzadeh & Emadi (2024) using weight prediction error and super-directive beamformer scoring 1.01 with SNR ranging between 0 and 20 dB.

Another high score was reported by Liu et al. (2023) obtaining 3.60 using text-informed speech enhancement for noise-robust keyword spotting (TE-KWS) used in the short-time Fourier transform (STFT) domain. The article employed the VCTK database (Veaux, Yamagishi & King, 2013) and Ego noise at SNR = −4 dB. Using Mel-FilterBank features with CLDNN and super-directive beamforming for speech enhancement (Olivieri et al., 2023) scored 1.73. The article used an In-house database with environmental noise for evaluation.

Just like PESQ, a higher STOI value indicates better performance. A total of six articles reported STOI scores as shown in Fig. 9. For the CHiME database, Shimada et al. (2019) scored 0.94. Using LibriSpeech database, Velásquez-Martínez et al. (2023) scored 0.84. For the TIMIT database (Zue, Seneff & Glass, 1990) using a super-directive beamformer in a reverberant environment (Shukla, Shajin & Rajendran, 2024) scored 0.96 with the SNR ranging between 5 and 25 dB. The lowest STOI average was reported by Jukić, Balam & Ginsburg (2023), scoring 0.70 with the WSJ database. The best STOI score was obtained by Liu et al. (2023) obtaining an average of 0.98.

Figure 10 compares the ASR performance in various types of noises, including ego noise, environmental noise, and reverberation, and the combinations of these noises that have been occurring in several studies, such as “Environmental + Reverberation.” Many studies on environmental noises in various scenarios presented separate results for each type of noise (Kim et al., 2012; Meyer et al., 2016; Sainath et al., 2017; Shimada et al., 2019; Yadava & Jayanna, 2019; Purushothaman, Sreeram & Ganapathy, 2020; Bu et al., 2022; Chai et al., 2022; Deepak et al., 2022; Velásquez-Martínez et al., 2023; Narayanan et al., 2023; Chang et al., 2023; Qu, Weber & Wermter, 2023; Olivieri et al., 2023; Lin et al., 2023; Shukla, Shajin & Rajendran, 2024; Cherukuru & Mustafa, 2024).

Figure 10 Average WERs for types of noise.

Though a comparative analysis of these methods was not possible due to a varying number of scenarios mentioned in each article at different SNR levels, the average of these values at each SNR level was incorporated in the results. It was observed that the average WER for the articles with environmental noise and reverberation was the highest at 24.77% followed by environmental noise at 20.56%. While the best score was achieved in non-stationary Environmental + Ego noise at 4.52%.

The WER value for reverberation and ego noise was recorded at 14.77%, which used a hybrid HMM-DNN with a delay-and-sum beamformer (Novoa et al., 2021). Environmental noise and ego noise had the highest WER value of 59.5% using the variational auto-encoder (VAE) and NMF (Fang et al., 2023). The environmental noise and reverberation category had an average WER of 24.78% in Sainath et al. (2017), Olivieri et al. (2023) which uses Mel-FilterBank features with CLDNN (Olivieri et al., 2023) and MVDR beamformer (Sainath et al., 2017). Environmental noise had an overall average WER of 23.96%, which was comparatively higher than most of the categories in Moritz, Anemuller & Kollmeier (2015), Qu, Weber & Wermter (2023), Cherukuru & Mustafa (2024). Finally, the ego noise category had a low score of 6.12% in Shukla, Shajin & Rajendran (2024) using BRO-DNN and low pass Filter.

RQ3- What are the challenges, limitations, and future research directions raised by the previous studies in multichannel speech enhancement for automatic speech recognition?

Table 5 depicts the limitations that need to be addressed in the future. Limitations related to resources were discussed in eight articles (Purushothaman, Sreeram & Ganapathy, 2020; Novoa et al., 2021; Huang et al., 2022; Qu, Weber & Wermter, 2023; Kanda et al., 2023; Dimitriadis et al., 2023; Lin et al., 2023; Cherukuru & Mustafa, 2024), with most of them highlighting the lack of real-world data (Qu, Weber & Wermter, 2023; Kanda et al., 2023; Cherukuru & Mustafa, 2024). For example, Cherukuru & Mustafa (2024) highlights the need for more extensive and varied datasets to understand better the system being developed. Huang et al. (2022) discusses how clustering-based methods using overlapping speech scenarios struggle in real-world situations and the need to train the proposed system on a large set of real-world and simulated data, while Qu, Weber & Wermter (2023) mention the lack of real-world speech data for evaluation. The articles mentioned above do not clearly define the properties of the “real-world data” they refer to, aside from a few minor indications. One such indication is the mention of it being spontaneous (Qu, Weber & Wermter, 2023), suggesting that the characteristics of background noise can vary across different environments. Additionally, when it comes to speakers, factors such as tone and speech rate may vary across different sentences. Overall, the signal should include realistic and diverse combinations of noise and speech.

Table 5 The limitations that need to be addressed in the future.

Area	Articles	N	
Resources	Kanda et al. (2023), Dimitriadis et al. (2023), Purushothaman, Sreeram & Ganapathy (2020), Qu, Weber & Wermter (2023), Huang et al. (2022), Cherukuru & Mustafa (2024), Lin et al. (2023), Novoa et al. (2021)	8	
Computational complexity	Sainath et al. (2017), Dua, Aggarwal & Biswas, 2019b, Subramanian et al. (2022), Shi et al. (2022), Shukla, Shajin & Rajendran (2024)	5	
Algorithms, models	Dua, Aggarwal & Biswas, 2019b, Velásquez-Martínez et al. (2023), Park et al. (2023), Kim et al. (2012), Velásquez-Martínez et al. (2023), Wang & Cavallaro (2021), Liu et al. (2023)	7	
Performance metrics	Jiang et al. (2023), Qu, Weber & Wermter (2023), Jokisch et al. (2021)	3	
Methodology (comparability, scope, continuation	Narayanan et al. (2023), Velásquez-Martínez et al. (2023), Chhetri et al. (2018), Velásquez-Martínez et al. (2023), Moritz, Kollmeier & Anemuller (2016), Dua, Aggarwal & Biswas, 2019a, Pandey et al. (2021), Chai et al. (2022), Subramanian et al. (2022), Huang et al. (2022), Velásquez-Martínez et al. (2023), Lin et al. (2023), Fang et al. (2023) Wang & Cavallaro (2021)	14	

Computational complexity was raised in five articles (Sainath et al., 2017; Subramanian et al., 2022; Shi et al., 2022; Jukić, Balam & Ginsburg, 2023; Shukla, Shajin & Rajendran, 2024); for example, the complexity of an increase in window size (Sainath et al., 2017). In Shukla, Shajin & Rajendran (2024), computational complexity affects the implementation of the proposed system in practical applications.

Limitations of the methodology were discussed in 14 articles. The lack of generalized solutions in data and environments made the proposed solutions viable for specific scenarios. For example, Dua, Aggarwal & Biswas (2019a) pointed out the computational expense and poor generalization of the maximum mutual information (MMI) techniques for unseen data, while (Jiang et al., 2023), the deep CNN (DCNN) model showed poor generalization abilities with changing data. The next limitations were computational complexity and hardware limitations (Sainath et al., 2017; Dua, Aggarwal & Biswas, 2019a; Subramanian et al., 2022; Shi et al., 2022; Shukla, Shajin & Rajendran, 2024).

In terms of potential future works, it includes the improvement in robustness and accuracy of the proposed models (Chhetri et al., 2018; Shimada et al., 2019; Lin et al., 2023), the investigation of alternative speech enhancement schemes and models (Ochiai, Watanabe & Katagiri, 2017; Jukić, Balam & Ginsburg, 2023), increasing the size and diversity of the datasets (Dimitriadis et al., 2023; Cherukuru & Mustafa, 2024), the reduction of model complexity to reduce training time with an increase in inference deployment (Pandey et al., 2021; Shukla, Shajin & Rajendran, 2024), increasing model size (Narayanan et al., 2023; Park et al., 2023; Dimitriadis et al., 2023), evaluating proposed methods in more challenging and realistic environments (Subramanian et al., 2022; Shi et al., 2022; Liu et al., 2023; Sadeghi, Sheikhzadeh & Emadi, 2024; Shukla, Shajin & Rajendran, 2024), addressing the computational and implementational challenges (Shukla, Shajin & Rajendran, 2024), optimization of test setup (Jokisch et al., 2021), and investigation of the proposed method performance on different drone platforms (Jokisch et al., 2021).

Discussion

This section discusses the significant findings from constructively analyzing and synthesizing data from the selected articles.

RQ1-The existing approaches in MCSE

Beamforming is one of the most common approaches used in multichannel speech enhancement due to its usability for several purposes like noise suppression (Li et al., 2021), dereverberation (Chai et al., 2022), and estimation of direction of arrival (DOA) (Sainath et al., 2017; Li et al., 2021; Chai et al., 2022; Subramanian et al., 2022; Olivieri et al., 2023; Dimitriadis et al., 2023). Beamforming is favored among researchers as it offers a variety of techniques like the neural beamformer (Ochiai, Watanabe & Katagiri, 2017; Purushothaman, Sreeram & Ganapathy, 2020; Shi et al., 2022; Chang et al., 2023; Jukić, Balam & Ginsburg, 2023), MVDR beamformer (Sainath et al., 2017; Chhetri et al., 2018; Li et al., 2021; Bu et al., 2022; Chai et al., 2022; Subramanian et al., 2022; Park et al., 2023), and the simple delay-and-sum beamformer (Novoa et al., 2021; Sadeghi, Sheikhzadeh & Emadi, 2024), enabling researchers to pick a specific beamformer that suits their requirements. From the review of MCSE techniques, we found that the MVDR (Sainath et al., 2017; Chhetri et al., 2018; Li et al., 2021; Bu et al., 2022; Chai et al., 2022; Cherukuru, Mustafa & Subramaniam, 2022; Park et al., 2023) is one of the most common beamforming techniques among the studies that use spatial information, which is better than its contemporary delay and sum (DS) beamformer, as shown in Chhetri et al. (2018) that utilizes DS beamformers, which struggles with echo paths not being determined uniquely (Zhang & Wang, 2021), causing the WER to increase in the noisy and reverberant environment (Chhetri et al., 2018). On the other hand, other signal processing methods like NMF were not used as numerously as beamforming. NMF was used only in three articles due to its limitations, including the need for iterative optimization (Shimada et al., 2019), which makes it complex and less suitable for real-time applications and is suitable in more specialized applications utilizing spectral decompositions. In short, beamformers can perform more numerous and a variety of tasks with less resource intensiveness compared to NMF; therefore, researchers have opted for them more often.

For MCSE, DNN/CNN-based MCSE algorithms was reported to have higher accuracy compared to traditional MCSE techniques (Pandey et al., 2021; Li et al., 2024) like spectral subtraction and can be trained to handle a variety of noise types, whereas spectral suppression despite being simple and easy to implement has certain limitations such as its static nature of denoising involves direct subtraction of noise spectrum from the signal (El-Fattah et al., 2014; Xia & Bao, 2014; Yadava & Jayanna, 2019), causing accuracy issues in non-stationary noisy conditions. Most of the studies utilized neural-network-based methods for the speech enhancement component in their methodology. This component involved methods of enhancing the speech signal by reducing SNR and noise filtering, among which DNN and CNN-based enhancement were common, while filters like the Kalman filter (Shukla, Shajin & Rajendran, 2024), Notch, and Lowpass filter (Jokisch et al., 2021) were used for signal denoising. Spatial processing is another type of speech enhancement involving various beamforming methods. MVDR Beamformer was used in most of the articles, while others opted for MCAEC and the spatial HuBERT.

For performance measures, WER is the most common performance measure on existing works, as the main objective of MCSE is to reduce the effect of noise in speech recognition. However, a handful of articles use PESQ and STOI as the performance measures metric. Among them, PESQ was applied in more research than STOI. STOI is the measure of speech intelligibility in the speech enhancement domain, and when compared with PESQ, STOI is sometimes replaced by other intelligibility methods like Mean Opinion Score and extended STOI (ESTOI) if greater accuracy is needed. Although ESTOI is more rigorous and accurate in measuring intelligibility, these methods require clean speech as a reference for measurement which may not be available outside of simulated scenarios.

RQ2- The current performance of MCSE in the existing studies

The highest WER score overall was obtained by Fang et al. (2023) 59.54%, which was attributed to the complexity of the noise, combining ego and environmental noise. Another factor for such a high WER value is that the results were obtained from the average of values across various SNR values. The best average WER of 4.52% was obtained by Pandey et al. (2021) using DCRN for speech enhancement. This approach applies complex spectral mapping, which jointly predicts real and imaginary components of speech. Additionally, it incorporates KL divergence to help the RNN-T-based ASR adapt to variations in speech signals. As a result, this method demonstrates cross-corpus generalization (Pandey & Wang, 2020) and superior WER results.

It was found that a combination of several noise types can impact the WER. For example, the highest WER of 59.5% was reported when recognizing speech from environmental and ego noise (Fang et al., 2023). The high WER indicates that simultaneously performing speech recognition in the presence of Ego and Environmental noises was a complex task. But at the same time, noise filtering in the preliminary stages of the recognition is known to have better results provided that the filtering process is carefully managed so that the creation of distortions is avoided (Braun & Gamper, 2022; Chang et al., 2023).

For environmental noise and reverberation, an average WER of 25.50 % was reported in Ochiai, Watanabe & Katagiri (2017), Cherukuru & Mustafa (2024), with the highest WER average of 38.81% (Cherukuru, Mustafa & Subramaniam, 2022) using DWT-CNN for the AURORA database. At the same time, Ochiai, Watanabe & Katagiri (2017) compare HMM-GMM and HMM-DNN using feature extraction methods such as LDA, TRAP, MFCC, and FBANK on the REVERB dataset.

For ego noise with reverberation, the average WER is 13.76%. in Novoa et al. (2021), a system that effectively manages listener movement, time-varying reverberation effects, environmental noise, and user position information for beamforming approaches in human-robot interaction (HRI) through a hybrid HMM-GMM-based ASR system. The proposed EbT-I system obtained a lower WER score than the other well-known third-party ASR tools, such as Google API, IBM API, and Bing API.

Conversely, the lowest average WER of 4.80% at SNR levels of 3 dB was reported in Jokisch et al. (2021) when recognizing speech in ego noise. For denoising, Jokisch et al. (2021) utilized a low pass filter, the Notch filter, and Google Cloud Speech-to-text API for speech recognition.

While combinations of two or more types of noise increase the WER, it does not necessarily mean that MCSE can result in low WER for single types of noise. For example, the reverberation category had an overall WER of 23.49%. In Chai et al. (2022), the neural network-based GSC (nnGSC) was experimented with using the AISHELL-2 clean corpus mixed with MUSAN noisy corpus at various SNRs, with a reported WER of 27.67%. However, the best average WER for reverberation was Sadeghi, Sheikhzadeh & Emadi (2024) with a WER of 9.15% when using a super directive beamformer and WPE for dereverberation on a spherical microphone array.

For environmental noise, it has an average WER of 15.76%. The highest average WER of 34.83% was recorded in Dua, Aggarwal & Biswas (2019b) using PSO-based feature optimization techniques, which have inherent issues like premature convergence to sub-optimal solutions, a common problem with metaheuristic optimization methods (Iinthapong et al., 2023). Cherukuru & Mustafa (2024) obtained an average WER of 26.17% on environmental noises from the AURORA database using beamforming, adaptive noise reduction, and voice activity detection. The proposed system employs DWT to denoise signals without compromising speech quality. The proposed CNN model shows considerable performance in recognizing speech under various environmental noises in stationary and non-stationary conditions.

Based on the results presented in the existing works, the WER is relatively low in a single type of noise scenario and much higher when involving two or more types of noise. This is expected as the complexity of dealing with several noise signals is non-trivial, and more effort is thus needed in this scenario.

In the case of PESQ, the lowest score of 1.01 was obtained by Sadeghi, Sheikhzadeh & Emadi (2024) aiming to improve WER in the ASR systems in noisy reverberant environments. This article employed WPE for dereverberation and a super-directive beamformer for speech enhancement in a highly reverberant environment with the RT60 up to 1,000 ms. The low score can be attributed to the late reverberation component mentioned in the article which can reduce speech quality and intelligibility, negatively impacting the PESQ scores. The best average PESQ and STOI scores of 3.60 and 0.98 were obtained by Liu et al. (2023), using TE-KWS with STFT for Ego Noise. TE-KWS provides robustness in keyword spotting in noisy environments, while STFT divides the signals into small segments of equal size and performs Fourier transform on them separately. Liu et al. (2023) attributes this score to the use of a tripartite branch structure consisting of a speech branch, alignment branch, and text branch which allows the model to effectively align text with speech. The lowest STOI scores of 0.70 were reported by Jukić, Balam & Ginsburg (2023) complemented by a good WER score of 14.12 using mask-based Neural beamforming with time-frequency masking. The low average in this case can be attributed to the cross-channel attention used in the flexible models, which does not significantly improve STOI scores compared to the models without it.

RQ3 Challenges, limitations, and future directions

One of the aims of this review is to examine the current challenges and limitations and possible future directions for overcoming those challenges and limitations.

Lack of interpretability

The issue of generalization and comparability was common among studies in the MCSE domain, affecting the different aspects of the proposed methods directly or indirectly. The systems or frameworks that work well under closed conditions cannot be adopted for practical applications. The failure of these systems for real-world applications was mainly due to the constraints of training data, models, or techniques. The CHiME-3 dataset was limited to real-world and far-field speech, which caused generalization issues. Similar dataset-related generalization issues were mentioned across multiple studies. This included being dependent on a single dataset for evaluation or an in-house dataset that may not represent the diversity of the real-world scenarios. It can be adapted to many different noise environments by combining more generalized approaches, which is very important for future research in the MCSE and ASR domains.

The deep convolutional neural network (DCNN) based acoustic model lacks interpretability, making it difficult for researchers to understand its decision-making due to its black-box nature. Besides being vulnerable to overfitting and adversarial attacks, DCNN also struggles with domain shift affecting its generalization in unseen domains. Jiang et al. (2023), Researchers need to reduce the interpretability gap of DCNN as a possible future direction.

Model size

The DNNs generally have good generalizability (Guan & Loew, 2020), yet some gaps need to be addressed. The gaps include the increase in model complexity as the learning becomes more challenging and the increase in training data size. The DNNs are optimized through mini-batch gradient descent algorithms for training large DNNs, which, while increasing the efficiency of the models, also causes a generalization gap (Oyedotun, Papadopoulos & Aouada, 2023). The future work must have a balance between the performance and model size to achieve an optimum solution.

Computational cost

Computational complexity and hardware limitations also limit the performance of MCSE-ASR. As the complexity of the models or the size of the data increases, more computational resources are needed. A major trade-off in MCSE lies between computational cost and the enhanced quality of the signal. Researchers also focus on optimizing MCSE algorithms to maximize the balance between performance and computational efficiency.

Deploying computer-intensive models in a resource-constrained environment is still challenging (Georgescu et al., 2021). Hardware limitations and computational constraints also limit the implementation of MCSE and ASR systems in real-world applications. The key constraints include computational power, memory usage, latency constraints, and energy/power efficiency (Georgescu et al., 2021). One possible future direction is minimizing the complexity of MCSE and ASR systems by incorporating big data management techniques that reduce computing requirements.

Performance metrics

The common issue with performance metrics is that they may not fully capture the subjective accuracy required by human listeners. A system with very low CER or WER may still not make sense to a human listener. Similarly, a signal with a good STOI or PESQ score may still be unnatural to a human listener. As such, there is a need for more effective performance measures that can be used as standard metrics for better comparability.

Future directions and key areas for improvement

The review provides us with several future directions to explore, such as conducting a technical review of the best-performing components of MCSE and ASR based on the results. Existing frameworks can be reviewed using a common dataset and experimental setup with good PESQ and WER scores. There also is a need for a generalized database for speech enhancement and recognition based on real-world speech and noise scenarios, which was lacking in the existing studies. Such a dataset can contain a variety of speech data like monologues, dialogues/discussions, formal and informal speech, and all the types of noise to simulate real-world noisy scenarios for the common types of speech. During the ICASSP 2023 Clarity Multichannel Hearing Aid Challenge, some works (Lei et al., 2023; Liu & Zhang, 2023) acknowledge the real-life application problem due to noise. Solutions such as Spectro-Spatial filters with neural networks (Lei et al., 2023; Liu & Zhang, 2023) show superior generalization performance on real-world data. Such solutions have potential for exploration in the future.

Figure 11 depicts the key areas that researchers should focus on in future research on MCSE for ASR systems.

Figure 11 Key areas that need attention in MCSE for ASR future research.

Multichannel speech enhancement leverages the advantages of spatial processing with the rapid growth in deep learning. Despite the rise and promising prospects of deep learning, some speech enhancement techniques are being neglected by researchers, particularly NMF. NMF is a strong speech enhancement technique applicable to both single-channel and multichannel SE and also offers feature extraction with good results. Similarly, CASA was utilized in the least number of articles, it is designed to replicate the human perception of sound segregation, making it highly effective in complex auditory environments.

Dependency on deep learning has limited the use of traditional speech enhancement techniques like spectral subtraction and signal compression algorithms like dynamic range compression (DRC). The DRC is a vital technique in scenarios where multiple sounds must be balanced to harmonize the listening experience, like in music, but in speech, it produces artifacts, and over-compression can lead to flat and unnatural speech. On the other hand, Spectral subtraction is an effective noise suppression technique that is simple to implement, efficient, and suitable for real-time applications. Therefore, using the latest deep learning-based techniques to complement the traditional techniques should be a focus in future research.

In terms of noise types, the datasets used in different articles incorporate only one or two types of noise to generate noisy data. Most studies do not point out the frequency of the noise or the stationary/non-stationary nature of the noise being simulated, which can cause inaccuracies in comparing experiments and the results of different articles.

As mentioned above, the articles use limited types of noise in many cases without reverberation; these scenarios may return good results but do not represent real-world noisy environments. In real-world scenarios, multiple types of noise may occur in the same environment at the same or varying frequencies, which is lacking in the articles reviewed.

In terms of performance, the high WER scores in the noisy low SNR scenarios could be caused by several possibilities, such as the noise-robust ASR systems’ less accurate performance at low SNR levels compared to high SNR. Another reason can be that the noise suppression algorithms in MCSE-ASR do not provide a clear enough output for the ASR to use for accurate results at low SNR. More focus is needed on increasing the recognition accuracy at low SNR.

As mentioned earlier, the high WER of several studies at low SNR levels points to the performance of the noise suppression or MCSE module. However, these factors cannot be established as the cause of deteriorating performance at low SNR because many articles in the study did not conduct performance tests for MCSE components separately. Therefore, we cannot calculate the impact of MCSE output on the accuracy of ASR. By including performance measures such as PESQ and STOI, it is easier for researchers to pinpoint the component that needs more attention in recognizing noisy speech.

Conclusions

To conclude, this SLR has extensively examined MCSE for ASR. The review highlighted the significant works and their methodologies, evaluation databases, noisy environments, and experimental setup in a detailed discussion. The review also highlighted the significant advancements and challenges in the domain of MCSE and ASR. Key findings include identifying effective MCSE methods that are helpful to ASR performance across various noisy environments. Future research should focus on optimizing MCSE-ASR frameworks for better performance in various real-world noisy scenarios, address the computational constraints, and explore new applications in diverse environments.

Supplemental Information

Supplemental Information 1 Quality Assessment Questions and Scores.

Supplemental Information 2 Performance of MCSE-ASR.

Additional Information and Declarations

Competing Interests

The authors declare that they have no competing interests.

Author Contributions

Zubair Zaland conceived and designed the experiments, performed the experiments, analyzed the data, performed the computation work, prepared figures and/or tables, authored or reviewed drafts of the article, and approved the final draft.

Mumtaz Begum Mustafa conceived and designed the experiments, performed the experiments, analyzed the data, performed the computation work, prepared figures and/or tables, authored or reviewed drafts of the article, and approved the final draft.

Miss Laiha Mat Kiah conceived and designed the experiments, performed the experiments, performed the computation work, authored or reviewed drafts of the article, and approved the final draft.

Hua-Nong Ting conceived and designed the experiments, performed the computation work, authored or reviewed drafts of the article, and approved the final draft.

Mansoor Ali Mohamed Yusoof conceived and designed the experiments, performed the experiments, analyzed the data, performed the computation work, prepared figures and/or tables, authored or reviewed drafts of the article, and approved the final draft.

Zuraidah Mohd Don conceived and designed the experiments, performed the computation work, authored or reviewed drafts of the article, and approved the final draft.

Saravanan Muthaiyah conceived and designed the experiments, performed the computation work, authored or reviewed drafts of the article, and approved the final draft.

Data Availability

The following information was supplied regarding data availability:

This is a literature review.

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
