# Peer review of "Multichannel speech enhancement for automatic speech recognition: a literature review"

_PeerJ Computer Science, doi:10.7717/peerj-cs.2772_

## Round 0.1 · original submission · Major Revisions

Dear authors,

Thank you for submitting your Literature Review article. Reviewers have now commented on your article and suggest major revisions. We do encourage you to address the concerns and criticisms of the reviewers and resubmit your article once you have updated it accordingly. When submitting the revised version of your article, it will be better to also address the following:

1. The Introduction section should contain a well-developed and supported argument that meets the goals set out.
2. How this review paper will contribute to the scientific body of knowledge should be clearly mentioned.
3. The coverage (both temporal and domain) of the literature and how the literature was distributed across time domains should be clearly provided.
4. Reviewer 1 & 2 have asked you to provide specific references. You are welcome to add them if you think they are relevant and useful. However, you are under no obligation to include them, and if you do not, it will not affect my decision.

Warm regards,

·

Basic reporting

This paper articulated the motivation for writing this review—the investigation of the impact of existing approaches in MCSE for ASR performance. The authors employed a scientific method to screen the approaches, ultimately identifying 40 sufficiently representative articles in the relevant field. By statistically analyzing their WER, PESQ, STOI, and other metrics at different signal-to-noise ratios, the authors illustrated the performance limits and trends of MCSE methods.

Numerous minor errors significantly reduce the readability and comfort. Writing details need careful revision, such as the MVDR acronym on line 248, which was not spelled out the first time it appeared but was spelled out the second time. The full form of NMF also appeared multiple times. Additionally, on line 796, there is 'Dynamic Range Compression (DCR)'??

Experimental design

It is hoped that the theoretical differences between the various methods can be supplemented, using formulas or diagrams to illustrate them. This would enhance the value and contribution of this paper, and also help newcomers to the field to better understand and get started.

Validity of the findings

There are some peculiar issues in the experimental section. I noticed that in Figure 4, there is a point with an SNR < 0 that achieved a very high PESQ score (around 3.6). Which method does this point correspond to, and why does it deviate so far from the linear function? If this deviated value and the 50 dB value (several figures do not specify the unit dB for SNR) are removed, the figure would yield a result completely opposite to the conclusion stated in your text.
1. Indicate the method corresponding to each point in Figures 3, 4, and 5;
2. Analyze the reasons for the deviation.

Additionally, the paper has already mentioned the issue of the lack of real-world scenario simulation, which is crucial for the practical application of MCSE. In last year's ICASSP 2023 Clarity Multichannel Hearing Aid Challenge (https://claritychallenge.org/docs/icassp2023/icassp2023_results), there was a case where a multichannel DNN end-to-end solution performed very well on simulated data but its scores dropped sharply when applied to real-world data. In such cases, solutions combining Neural Spectrospatial Filtering and DNN showed better generalization performance, achieving first place in the real-world data track (Eval2). It would be beneficial for the authors to discuss this in the paper.

Additional comments

Would it be more appropriate to replace the last keyword 'Noise' with 'Noise Types' or 'Noise Scenarios'?

For some literature references, such as 'Flexible Multichannel Speech Enhancement for Noise Robust Frontend,' if there is an IEEE citation available, please use that instead of arXiv.

Reviewer 2 ·

Basic reporting

1. One major issue in the main body (Lines 219-823) is that the authors have used too many inappropriate and ambiguous statements, which are caused by either poorly-designed method categorization or factual errors.
1) Beamforming actually belongs to spatial filtering techniques. So the current terminology should be refined. Based on the popularity of signal-processing-based MCSE techniques, I would recommend dividing them into the following categories: Beamforming, Blind Source Separation (BSS), multichannel NMF, and Computational Auditory Scene Analysis (CASA). So I think the “Multichannel Speech Enhancement” section requires a major revision.
2) The authors should note that environmental noises are generally not white or colored. They have complex frequency components that usually cannot be fully modeled by the simple assumptions in white noise or colored noise. In the current manuscript (e.g., Lines 344, 652-654), however, the authors seem to categorize environmental noise as white or colored noises.
3) In Line 237, this statement is inaccurate, because the beamforming performance can deteriorate significantly as the reverberation time increases. So beamforming is often combined with other techniques (e.g., WPE) to achieve joint denoising and dereverberation.
4) In Line 242, the distance information may not have to be known in fixed beamforming.
5) In Line 254, VAE is not directly used for filtering in the referenced paper. It is instead used for speech modeling, while the actual denoising is achieved by multichannel Wiener filtering.
6) In Lines 256-257, VarArray is a purely neural network based MCSE approach, which I don't think can be categorized as a spatial filtering method.
7) In Line 270, to avoid confusion with the previous section, it is recommended to rename this section. Maybe “Single-channel Speech Enhancement” is better.
8) There is no clear correspondence between the section titles and the “Types” column in Table 3. In addition, the rows in the “Type” column in Table 3 are not well designed, as their names seem to be very similar and ambiguous (e.g., Spatial Processing vs. Spatial Processing & Speech Enhancement vs. Speech Enhancement). Speech enhancement can be a very broad concept which may include spatial processing and other techniques in my understanding. So it is better to redesign the table. One possibility is to divide the methods into two categories: signal processing based and purely neural network based. This also applies to Figure 7.
9) Similarly, in Line 275, "noise suppression" can also have a broad definition that covers various types of techniques (including MCSE methods). If the authors cannot find a more precise name for this line of research, it is better to re-organize the paragraphs and merge the methods to other categories.
10) In Line 355-259 and Table 4, Librispeech and TIMIT are mostly clean speech with very limited background noise. In the related SE studies, it is often further mixed with additional noise samples to form a noisy speech dataset. DEMAND is only a multichannel noise corpus, so it is not associated with any language. It is better to indicate the data type of each row in Table 4, e.g., noise, clean speech, noisy speech. In addition, it would be helpful to add the publication year of each dataset.
11) In Lines 402-407, the value range of STOI should be mentioned.
12) Figure 3 should be mentioned at the beginning of RQ2 (Lines 426-430). Otherwise it is difficult for readers to follow the analysis in different SNR conditions.
13) In Line 662, the statement is inappropriate as ESTOI is not computationally heavy.
2. A thorough grammar and spelling check is required to improve the English usage in this paper, as I have found many similar issues.
1) "MSCE" -> "MCSE"
2) "(3) discusses" -> "(3) discuss the"
3) "provide (1) a" -> "(1) provide a"
4) "two basic types fixed and" -> "two basic types: fixed and"
5) "array-geometry-agnostic" -> "Array-geometry-agnostic"
6) "(Liu et al., 2023) The common techniques" -> "(Liu et al., 2023). The common techniques"
7) "(Fang et al., 2023)The average WER" -> "(Fang et al., 2023). The average WER"
8) "Figure 4 and Fig.5" -> "Figure 4 and Figure 5"
9) "but in speech. It" -> "but in speech, it"
10) Line 800-801 is an incomplete sentence.
3. The figures should be improved to have a higher quality (DPI) and clarity.
4. Finally, there are some reference-related issues:
1) A tutorial article [R1] should be compared and discussed as it also includes a comprehensive review of representative MCSE techniques used for far-field ASR.
2) When introducing neural beamformers, pioneer works should be referenced, such as [R2], [R3], and [R4].
3) In Lines 242-244, the references are inappropriate when referring to fixed beamforming and adaptive beamforming. They should be replaced with representative articles or surveys on the corresponding topic.
4) References to the original papers should be added when introducing PESQ, STOI, and other metrics in Lines 390-394.
5) Some reference notations are ambiguous and it is difficult to find the corresponding articles. For example, “Lin et al., 2023” can refer to either Lines 944-946 or Lines 947-948.

[R1] R. Haeb-Umbach, J. Heymann, L. Drude, S. Watanabe, M. Delcroix, and T. Nakatani, “Far-field automatic speech recognition,” Proceedings of the IEEE, vol. 109, no. 2, pp. 124–148, 2021.
[R2] H. Erdogan, J. R. Hershey, S. Watanabe, M. I. Mandel, and J. Le Roux, “Improved MVDR beamforming using single-channel mask prediction networks,” in Proc. ISCA Interspeech, 2016, pp. 1981–1985.
[R3] J. Heymann, L. Drude, and R. Haeb-Umbach, “Neural network based spectral mask estimation for acoustic beamforming,” in Proc. IEEE ICASSP, 2016, pp. 196–200.
[R4] X. Xiao, C. Xu, Z. Zhang, S. Zhao, S. Sun, S. Watanabe, L. Wang, L. Xie, D. L. Jones, E. S. Chng, and H. Li, “A study of learning based beamforming methods for speech recognition,” in CHiME 2016 workshop, 2016, pp. 26–31.

Experimental design

The research questions are reasonable and meaningful. The corresponding search and filtering strategies for collecting related articles also look satisfactory. According to Figure 2, the finally selected articles comprise both recent studies (2020~2024) and older ones (since 2014), with most articles published in the past 5 years. This should form a good basis for the review.

However, the methodology for experimental result analysis inappropriate in my opinion.
1) In Lines 383-407, it does not make much sense to me to compare WERs from different articles if they use very different evaluation data and ASR systems. It is recommended to re-calculate the average WERs at least based on the same database.
2) Similarly, it is less informative to simply compare the metrics of different papers in Table 5 and Lines 383-823, as they may use very different evaluation data. It would be clearer if the articles in Table 5 can be grouped according to the evaluation data they used. It is also meaningless to average the performance across different articles as they have used different evaluation data.
3) It is not informative either to look at individual points in Figure 3 where the WER is the lowest or highest without making sure the same data setup is adopted. Because poor WERs can be caused by both difficult acoustic conditions (e.g., low SNRs) and weak ASR systems (e.g., small models with limited training data). It would be more interesting to demonstrate the performance evolution on the same dataset in different years.
4) The corresponding conclusions (e.g., Lines 466-467, 667) are also inappropriate as the results are simply not comparable due to data and model discrepancies in different articles.
Since the aforementioned weird methodology is adopted throughout the paper, a significant revision and redesign of the result analysis is required.

Validity of the findings

As mentioned in the "Study design" part, the methodology for experimental result analysis is problematic and requires a significant revision. Therefore, the corresponding findings/conclusions obtained from such an analysis are not sufficiently convincing or informative.

Additional comments

1. In the "Performance measure" section (Lines 383-407), it is better to mention the availability of different metrics. For example, WER, CER, and so on can be used when the corresponding transcript is available, which do not require a clean reference speech label. In contrast, PESQ, STOI, and so on can be used only when the clean reference speech is available, which often reply on simulation data.
2. In Lines 574-575, it is better to elaborate on the specific types of real-world data needed, e.g. diverse array geometries, realistic environments, diverse speaker traits.
3. The summary of RQ2 in the “Discussion” section (Lines 666-718) should be more concise and conclusive. Currently it looks very similar to the descriptions in Lines 383-566. In addition, the analysis methodology should be revised (also mentioned above), as results from different articles usually cannot be compared or averaged unless they share the same evaluation data.
4. In Lines 735-743, although much space is used to describe the DCNN method, the DCCN based acoustic model is not a popular architecture in modern ASR systems. The related references should also be updated to reflect the latest tendency in the literature.

---

## Round 0.2 · Minor Revisions

Dear Authors,

Thank you for submitting your revised article. Reviewers have now commented on your study. Both reviewers think that your article need minor revision, and it has still not been recommended for publication in its current form. However, we encourage you to address the concerns and criticisms of reviewers and to resubmit your article once you have updated it accordingly.

Best wishes,

·

Basic reporting

The revised paper provides sufficient information for researchers and developers with the foundational knowledge of current advancements and limitations in the field of MCSE for ASR.

Experimental design

no comment

Validity of the findings

no comment

Additional comments

1. The formatting of the references is inconsistent; for instance, several articles from the ICASSP conference are cited in varying formats. Furthermore, some references mentioned in the paper lack corresponding entries in the reference list, such as (Liu & Zhang, 2023). If you are composing the article in Word, I recommend utilizing reference management tools like Zotero or EndNote. This will also enhance the functionality of clickable references.

2. The number of effective bits in Figure 10 should be standardized for consistency.

Reviewer 2 ·

Basic reporting

1. Most of the inappropriate and ambiguous statements pointed out in the last round of review have been addressed. However, some incorrect claims or ambiguity still exist in the revised manuscript.
1) The guided source separation (GSS) method, as indicated by its name, requires prior knowledge about the sources (e.g., source activity information) to achieve accurate separation. It cannot be categorized as BSS techniques.
2) The sentence "Metrics like WER, CER, and SER are independent of corresponding data." is not clear enough, which should be rephrased.
2. Some grammar issues or typos need to be addressed.
1) Lines 264-266 should be split into two sentences.
2) "evalusation" -> "evaluation"
3) (Line 494) "2.64. Whereas" -> "2.64, whereas"
4) Lines 562-565 are not a complete sentence.
3. The figures should be further improved to have a higher quality (DPI) and clarity.
4. The precision of different results should be unified in Figure 10.
5. Finally, there are some reference-related issues:
1) Some terms are not correctly capitalized in the title, e.g., "Mandarin", "VarArray", "t-SOT", "Unet", and "TIMIT".
2) In Lines 1081-1082, the publication information is missing (i.e., CHiME 2016 workshop).

Experimental design

The major concern from the last round was the inappropriate methodology for experimental result analysis, which has been addressed in this revision. The experimental results are now grouped by the source corpora, making them much clearer to interpret. The corresponding conclusions are also more reasonable and useful.

Validity of the findings

Since the major concern regarding the study design has been addressed, the findings are now more sensible and convincing. I just have one major question in Figure 7:
* How is the number 26.18 obtained for CNN, DWT, Beamforming (Cherukuru and Mustafa 2024) in Figure 7? This number cannot be found in the referred paper. In addition, it is weird to see the paper published in 2024 shows much worse ASR results than the paper published in 2014 on the Aurora dataset.

---

## Round 0.3 · Minor Revisions

Dear Authors,

We will be awaiting the submission of the revised manuscript, with the requisite minor edits applied, in accordance with the feedback provided by two reviewers.

Best wishes,

·

Basic reporting

The paper provides a systematic literature review of multichannel speech enhancement (MCSE) techniques aimed at improving the robustness and accuracy of ASR systems. The review addresses the gap in existing literature by offering a comprehensive analysis of various MCSE approaches, models, and datasets, and their performance in different noise environments. The authors conducted an extensive search across multiple electronic databases, ultimately selecting 40 high-quality articles for detailed analysis. Key findings highlight the increasing trend in MCSE research, common performance measures such as WER and STOI and the challenges in achieving generality and comparability across different MCSE works. The review also identifies promising areas for future research, emphasizing the potential for further advancements in MCSE techniques to enhance ASR performance.

Experimental design

The paper's survey methodology is consistent with a comprehensive and unbiased coverage of the subject, ensuring a thorough examination of multichannel speech enhancement for ASR systems. Additionally, sources are adequately cited, and the review is logically organized into coherent paragraphs and subsections, facilitating a clear understanding of the topic.

Validity of the findings

Sufficiently adequate and credible.

Additional comments

1. Split lengthy sentences (e.g., Lines 52-56) for improved readability.
2. Ensure uniformity in the font of numbers in the images (e.g., the mixed use of Times New Roman and other fonts in Figures 9 and 10).
3. Change "several previous papers have attempted to summarize [...] but lack the accumulation and analysis" to “"existing surveys inadequately address critical aspects such as [...]"”.

Reviewer 2 ·

Basic reporting

The paper has been much improved compared to the initial version, which is now almost in good shape for publication. However, the following minor comment should be addressed.
* (Lines 265-266) typo: "This approach help" -> "This approach helps"

Experimental design

no comment

Validity of the findings

The findings are now better organized and presented. I just have some minor comments for improving the clarity.
* If the same averaging strategy as described in the authors' response is applied when calculating the WERs (and other metrics) from different papers in Figure 7, it would be helpful to mention it in the main text. This would help readers understand the results better.
* As explained in the authors’ response, the significant performance gap between (Cherukuru and Mustafa 2024) and (Martinez, Moritz, and Meyer 2014) on the Aurora data can be caused by multifaceted factors such as SNR settings, reverberations, and so on. Since this can also apply to many other comparisons throughout the paper, it is better to explicitly add related descriptions during analysis so that readers will not be confused by simply looking at the results. It would be even better if the difference of data configurations in different papers could be highlighted in the figures.

---

## Round 0.4 · accepted · Accept

Dear Authors,

Thank you for addressing the reviewers' comments. Your manuscript now seems sufficiently improved and ready for publication.

Best wishes,

[Reviewer 2 ·

Basic reporting

The paper is now in good shape after addressing all my concerns in previous review rounds.

Experimental design

no comment

Validity of the findings

The experimental analysis now looks much clearer thanks to the addition of SNR conditions and highlights of different data setups. The findings are sound and can be a good reference for practitioners in the speech enhancement field.